

# Factors influencing intention to intervene in elder abuse among nursing students

Hee-Jeong Kim[1], Min-Sook Seo[2] and Dahye Park[3]

[1] Department of Nursing, Namseoul University, Cheonan, Chungcheongnam-do, Korea
[2] Department of Nursing, Sangji University, Wonju-si, Gangwon-do, Korea
[3] Department of Nursing, Semyung University, Jecheon-si, Chungcheongbuk-do, Korea

## ABSTRACT

**Purpose**. This study aimed to identify the factors influencing the intention to intervene in elder abuse among nursing students.

**Methods**. A descriptive survey design was used, and questionnaires were completed by 182 nursing students. Statistical analysis was performed on the data collected, using SPSS 25.0, $\chi^2$-test, $t$-test, correlation, and regression analysis to confirm predictors of intention to intervene in elder abuse.

**Results**. Awareness of abuse, legal and institutional knowledge, and attitude were positively correlated with the intention to intervene in elder abuse. Factors influencing the intention to intervene by nursing students were education courses about elder abuse, awareness of elder abuse, legal and institutional knowledge, and attitude.

**Conclusion**. The findings suggest that it is necessary to provide educational protocols for nursing students to improve their awareness and knowledge of elder abuse. Based on the findings of this study, there is a need for specific education programs and guidelines to increase the assessment of, and intervention in, elder abuse.

## INTRODUCTION

In Korea, as of 2020, the population of older people, aged 65 or older, is 15.7% of the total population, and by 2040, it is predicted that 1/3 of the total population will consist of individuals aged 65 or older (*Statistics Korea, 2020*). The increase in the older people population has caused social problems beyond individual problems. Among them, abuse occurs in one in six older people worldwide, and it is becoming a serious problem due to the increase in the number of cases, and various aspects including depression. (*Yon et al., 2017*). Elder abuse is a concept that is influenced by the culture and values of society and includes not only physical abuse, but also emotional and sexual abuse (*Hall, Karch & Crosby, 2016*; *Lachs & Pillemer, 2015*). According to a press release on elder abuse statistics in 2018, the number of abuse cases in Korea increased by 12.2%, from 4,622 cases in 2017 to 5,188 cases in 2018 (*Korea Institute for Health and Social Affairs, 2019*).

In addition, according to the 2019 Elder Abuse Status Report, the number of reports of elder abuse reported by 34 regional older people protection agencies nationwide in Korea was 16,071, while the number of reports obligated to report elder abuse by mandatory reporters was only 877 (*Korea Institute for Health and Social Affairs, 2018*).

Corresponding author
Dahye Park, dhpark@semyung.ac.kr

Even considering the difference between the experiences of abuse and reporting, there are few reports by persons liable to report abuse of senior citizens. There are differences in the perception of elder abuse by senior citizens who experience abuse and those who are obliged to report it.

Elder abuse is not an individual or family problem, but a grave issue that requires social intervention and response, especially by nurses, who are the first medical personnel to attend to older people. Medical institutions, home care, and visiting nursing are accessible to the target person and are responsible for observing their physical and psychological condition (*World Health Organization, 2015*). Therefore, nurses should perform prevention and intervention tasks with an awareness of elder abuse and an attitude toward intervention. However, there were not many cases in which working nurses perceived behavior as elder abuse, and the number of reported cases was also very low, with 2.7% in 2019 and 3.4% in 2018 (*Korea Institute for Health and Social Affairs, 2019*; *Korea Institute for Health and Social Affairs, 2018*).

The intention of intervention in elder abuse includes activities such as educating the target person regarding elder abuse so that they do not suffer from the abuse again, establishing the status of the abuse, and informing the relevant institutions to provide care (*Stark, 2012*), which refers to the individual's willingness to stop the abuse of the suspect condition according to the nurse's own will (*Cho, 2014*). Previous studies have indicated that the variables that influence the intention of medical personnel to intervene in elder abuse were their attitudes toward intervention (*Cho, 2014*; *Feng & Wu, 2005*), abuse awareness (*Cho, 2014*; *Seong et al., 2016*), and legal and institutional knowledge (*Ko, 2010*). According to a study in Korea, research on elder abuse concerning nursing caregivers has been conducted, and the impact of clinical nurses' involvement in elder abuse has been reported (*Cho, 2014*; *Oh & Kang, 2016*), but research on elder abuse in the context of nursing students is rare. The education of undergraduate nursing students may become a future-oriented strategy for the long-term improvement of nurses' sensitivity and knowledge regarding elder abuse, as well as to improve the rate of abuse reports (*Lee & Kim, 2018*). Therefore, not only nurses, but also nursing students should be subject to systematic education regarding elder abuse. However, while Korea's undergraduate nursing programs meet the learning objectives regarding the care of older people, only a small portion of elder abuse-related content is covered. If an act of abuse is not recognized, it cannot lead to a report; therefore, it is important for nursing college students, who will come across cases of elder in the future, to grasp the perception of abuse. In addition, nursing students who will be part of the reporting duty group after graduation should receive systematic education on how to both judge and recognize elder abuse, and to be informed on the reporting system. For this, it will be necessary to identify and verify factors related to the intention to report elder abuse and to develop a systematic education program to strengthen vulnerabilities in the future. In response, we conducted this study to prepare basic data for creating awareness among nursing college students toward elder abuse by investigating the perceptions, knowledge, and attitudes of nursing college students in this regard.

## Purpose of research

This study aimed to identify the impact of nursing college students' perceptions of elder abuse, legal and institutional knowledge, and attitudes regarding the intention to intervene in elder abuse, and to use it as a basis for improving awareness of elder abuse and early detection.

The specific purpose of this study is as follows:

- To identify the differences in the intention of intervention in elder abuse according to the general characteristics of nursing college students.

- To verify the awareness of elder abuse, legal and institutional knowledge, attitudes of nursing college students, and their intention to intervene.

- To identify the factors that affect the intention of nursing college students to intervene in elder abuse

## MATERIALS & METHODS

### Research method

*1. Research design*

This study is a descriptive investigation to identify the impact of nursing college students' perceptions of elder abuse, legal and institutional knowledge, and attitudes regarding the intention of elder abuse intervention.

*2. Ethical consideration*

Ethical approval (IRB No: NSU-202005-004) was obtained from the Research Ethics Committee of Namseoul University. Participants were informed that they could leave the study at any time without penalty, and all personal information was kept confidential.

*3. Subjects of research*

The subjects of this study were 182 nursing college students from three nursing departments in Chungcheong-do and Gangwon-do, who understood the purpose of the study and agreed to participate. Using the G*power 3.1 program, the appropriate number of samples was at least 166, based on an effect size of 0.15, a power of 95%, and a significance level of 5%.

*4. Research variables*

*(1) Perception of elder abuse.* This study used the perception of elder abuse modified by *Kim & Lee (1998)* and *Jun & Song (1997)* and used the elder abuse recognition tool developed by *Lim (2001)* as 38 questions (*Kim & Lee, 1998*; *Jun & Song, 1997*; *Lim, 2001*). For the perception of elder abuse questionnaire, 38 items were rated on a 4-point Likert scale (1 = Strongly Disagree, 2 = Disagree, 3 = Agree, and 4 = Strongly Agree), that consisted of five subcategories: emotional abuse (10 items), verbal abuse (6 items), physical abuse (7 items), financial abuse (7 items), and neglect (8 items). Higher scores indicated worse elder abuse. When developing the instrument, in *Lim*'s (*2001*) study, the tool's reliability of subscales was emotional abuse .86, verbal abuse .79, physical abuse .84, financial abuse .89, neglect .86, and the overall reliability Cronbach's alpha was .92. In this study, the reliability of the instrument was $\alpha$ .97, and the reliability of the subscales was emotional abuse .86, verbal

abuse .91, physical abuse .98, financial abuse .92, neglect .86, and the overall reliability Cronbach's alpha was .92 (*Lim, 2001*).

*(2) Legal and institutional knowledge of elder abuse.* To measure legal and institutional knowledge of elder abuse, *Huh (2003)* revised and supplemented the measures used in the study on factors that affect the perception and reporting behavior of child abuse reporting obligators. The original scale consists of five items, but in this study questionnaire, there were 12 items that consisted of three categories: definition of concepts and types (four items), law of reporting persons and institutions (four items), and system of reporting agency and reporting number (four items). The participants were required to answer each question with "I don't know" and "I know." If they answered "I know," they would get one point. There were zero points for "I don't know." When the questions were added together, the score category ranged from 0 to 12. The combined score of 12 questions was the knowledge score for reporting elder abuse, and the higher the score, the higher the knowledge related to reporting. In this study, Cronbach's alpha of the tool was .66.

*(3) Elder abuse attitudes.* The attitude toward elder abuse tool was developed and used by *Park, Choi & Lee (2013)* in an intention to report study regarding a person liable to report child abuse, and *Cho (2014)* used a modified tool to suit older people. The tool consists of a total of 14 questions and is rated on a four-point Likert scale (1 = Strongly Disagree, 2 = Disagree, 3 = Agree, and 4 = Strongly Agree) that asks for the overall attitude toward intervention in elder abuse. Higher scores indicated more positive intervention behaviors. Examples of the questions include "If you report the elder abuse, appropriate measures will be taken by the reporting agency" and "If you intervene in the elder abuse, it will help them." In *Cho*'s (*2014*) study, Cronbach's alpha for the tool was .77, and in this study, the Cronbach's alpha for the tool was .68.

*(4) Intention to intervene in elder abuse.* The intention to intervene in elder abuse was a comparative study of perceptions and intention to report elder abuse between nurses and older people, used by *Ko (2010)* as a measurement tool to recognize elder abuse in her study on "Aware and Intent to Report Abuse of Older Adults." When asked if they would intervene in a hypothetical case with a total of 11 items, the scores for the four-point scale, where "Strong disagree" was 1 point, "Disagree" was 2 points, "Agree" was 3 points, and "Strong agree" was 4 points, were added together. Cronbach's alpha for the tool's reliability identified in the previous study was .79. In this study, Cronbach's alpha of the tool was .79.

### 5. Data collection

The data collection period was from July 9 to October 9, 2020. Recruited *via* a bulletin board, data were collected from 182 college students attending nursing departments at three universities in Chungcheong-do and Gangwon-do. The application form was set up beside the bulletin board for ease of application. The researcher and trained research assistant confirmed the eligibility of the participants, they were informed of the purpose of the study, and their consent was obtained by written. The invitation included a link that enabled the questionnaire to be activated, completed, and returned electronically. The questionnaire

was online and self-reported, and the participants were given a small gift to thank them for their participation. Response data were collected and stored using a questionnaire tool and subsequently exported for analysis. The questionnaire took approximately 15 min to complete.

### 6. Data analysis

The data collected in this study were processed using the SPSS 25.0, and the specific analysis methods were:

(1) The general characteristics of the subjects were calculated using frequency, percentage, average, and standard deviation using technical statistics, and the difference between the variables according to the general characteristics of the subjects was calculated using t-tests and $F$-tests.

(2) The subjects' perception of elder abuse, legal and institutional knowledge of elder abuse, elder abuse attitude, and intention to intervene in elder abuse were analyzed by means and standard deviation using technical statistics.

(3) Pearson's correction factors were analyzed to determine the correlation between variables.

(4) Multiple regression analysis was conducted to identify the factors affecting the intended elder abuse intervention.

## RESULTS

### Intent to intervene in elder abuse according to the general characteristics of the subject

A total of 163 women (89.3%) and 52 students (34.6%) from the fourth grade participated in this study. In this sample, 92 people (50.5%) said that they had no religion. In total, 162 respondents (88.5%) said they did not receive education on elder abuse, 179 (98.8%) said that they "agreed" with the elder abuse reporting law, and 2 (1.1%) said that they had experienced abuse. According to the analysis of the difference in the intent of the elder abuse intervention according to the general characteristics of the subject, the results indicated that education courses regarding elder abuse ($t = 3.067$, $p = .042$), elder abuse ($t = 3.333$, $p = .021$), elder abuse reporting ($t = 13.033$, $p < .001$), and the need for education ($t = 5.429$, $p = .005$), exhibited significant differences and did not differ in other variables (Table 1).

### Perception, legal and institutional knowledge, and attitude toward elder abuse and degree of intention to intervene in elder abuse

The average rating of the subject was 3.76 ±.33, out of a total of four points. Among them, the average rating of physical abuse was the highest level of recognition (3.95 ±.30), and emotional abuse was the lowest with an average rating of 3.51.40. Legal and institutional knowledge of elder abuse was 5.92 ± 2.29 points out of a total of 12, and attitudes toward elder abuse and intentions of elder abuse were 2.49 ±.33 points and 3.31 ±.41 points out of 4 points, respectively (Table 2).

**Table 1 General characteristics (N = 182).** Intent to intervene in elder abuse according to the general characteristics.

| Variables | Categories | n (%) | M ±SD | t/F | p |
|---|---|---|---|---|---|
| Sex | Men | 19(10.7) | 37.21 ± 4.41 | .553 | .458 |
| | Women | 163(89.3) | 36.38 ± 4.58 | | |
| Grade | 2 | 52(34.1) | 36.36 ± 4.56 | 3.342 | .711 |
| | 3 | 48(31.3) | 36.22 ± 4.87 | | |
| | 4 | 52(34.6) | 36.92 ± 4.00 | | |
| Religion | Buddhism | 15(8.2) | 35.86 ± 5.55 | .576 | .632 |
| | Christian | 56(30.8) | 37.10 ± 4.31 | | |
| | Catholic | 19(10.4) | 35.94 ± 4.93 | | |
| | No religion | 92(50.5) | 36.29 ± 4.50 | | |
| Education course about elder abuse | Has taken | 20(11.5) | 38.15 ± 4.18 | 3.067 | .042 |
| | Not taken | 162(88.5) | 36.26 ± 4.58 | | |
| Elder abuse problem | Not very serious | 2(1.2) | 39.00 ± 5.65 | 3.333 | .021 |
| | Not serious | 31(16.0) | 34.22 ± 4.92 | | |
| | Serious | 99(54.9) | 36.79 ± 4.28 | | |
| | Very serious | 50(28.0) | 37.12 ± 4.54 | | |
| Elder Abuse Reporting Act | Opposition | 3(1.2) | 27.33 ± 4.61 | 13.033 | .000 |
| | Agree | 179(98.8) | 36.62 ± 4.41 | | |
| Need for education | Unnecessary | 2(0.9) | 28.50 ± 9.19 | 5.429 | .005 |
| | Usually | 72(38.7) | 35.70 ± 4.05 | | |
| | Need | 108(60.4) | 37.12 ± 4.64 | | |
| Experienced elder abuse | Has taken | 2(1.1) | 35.50 ± 2.12 | .091 | .763 |
| | Not taken | 180(98.9) | 36.48 ± 4.58 | | |

**Table 2 Descriptive statistics of study variables.** Perception, legal and institutional knowledge, and attitude toward elder abuse and degree of intention to intervene in elder abuse.

| Variables | M ± SD | M ± SD | Min | Max | Range |
|---|---|---|---|---|---|
| Awareness of elder abuse | 3.76 ±.33 | 143.00 ± 12.81 | 38 | 152 | 38–152 |
| Psychological abuse | 3.51 ±.40 | 35.15 ± 4.01 | 10 | 40 | 10–40 |
| Verbal abuse | 3.85 ±.36 | 23.14 ± 2.21 | 6 | 24 | 6–24 |
| Physical abuse | 3.95 ±.30 | 27.70 ± 2.15 | 7 | 28 | 7–28 |
| Financial abuse | 3.79 ±.42 | 26.58 ± 2.95 | 7 | 28 | 7–28 |
| Elder neglect/ Elder dereliction | 3.80 ±.36 | 30.40 ± 2.95 | 8 | 32 | 8–32 |
| Knowledge | .49 ±.19 | 5.92 ± 2.29 | 0 | 12 | 0–12 |
| Justice | .67 ±.25 | 2.71 ± 1.02 | 0 | 4 | 0–4 |
| Law | .62 ±.25 | 2.49 ± 1.00 | 0 | 4 | 0–4 |
| System | .17 ±.23 | 0.71 ± 0.93 | 0 | 4 | 0–4 |
| Attitude | 2.49 ±.33 | 34.98 ± 4.71 | 26 | 52 | 1–56 |
| Intention of intervening in elder abuse | 3.31 ±.41 | 36.47 ± 4.56 | 22 | 44 | 11–44 |

## Correlation between perceptions, legal and institutional knowledge, and attitudes toward elder abuse and intention to intervene in elder abuse

The intention to intervene showed a significant positive correlation with the perception of elder abuse ($r = .26$, $p = .005$), legal and institutional knowledge ($r = .54$, $p = .046$), and attitude ($r = .121$, $p = .004$). Attitudes toward elder abuse showed a significant positive correlation with legal and institutional knowledge ($r = .15$, $p = .044$) (Table 3).

**Table 3  Correlations among the variables.** Correlation between perceptions, legal and institutional knowledge, and attitudes toward elder abuse and intention to intervene in elder abuse.

| Variables | 1 | 2 | 3 | 4 |
|---|---|---|---|---|
| | | r(p) | | |
| Awareness of elder abuse | 1 | | | |
| Knowledge | .011(.886) | 1 | | |
| Attitude | .071(.338) | .150(.044)* | 1 | |
| Intention of intervening towards elder abuse | .206(.005)* | .540(.046)* | .121(.004)* | 1 |

Notes.

*$p < .05$.
**$p < .01$.
***$p < .001$.

## Factors influencing the intention of nursing college students to intervene in elder abuse

Multiple regression analyses were conducted to determine the factors affecting the intended elderly abuse intervention. In this study, we used sex as a control variable, which showed a significant difference in the general characteristics of the intentions toward elder abuse intervention. Multiple regression analysis was conducted with the three main variables of this study (elder abuse awareness, legal and institutional knowledge of elder abuse, and attitudes toward elder abuse) that were independent variables, and the intention of elder abuse intervention was a dependent variable. Testing the assumptions required for the multiple regression analyses were all satisfied. The dependent variables showed a tendency for normal distribution, and residual analysis confirmed linearity, normality, and equivalence. Furthermore, the Durbin-Watson statistic was 2.12, and there was no autocorrelation between the error terms. Next, using tolerance and variance inflation factor (VIF), the presence or absence of multicollinearity between independent variables was found to be less than 1.0 from 0.73 to 0.95, and the VIF from 1.04 to 1.36, indicating that it was below the multicollinearity criterion of 10.0.

Finally, the regression analysis showed that the regression model was significant ($F = 4.349$, $p = .001$), representing 12.9% of the total explanatory power. The factors affecting the intent of elder abuse intervention were education on elder abuse ($\beta = 2.490$, $p = .035$), awareness of elder abuse ($\beta = 2.479$, $p = .011$), legal and institutional knowledge ($\beta = .054$, $p = .044$), and attitude ($\beta = .068$, $p = .029$), respectively. In other words, the higher the intention of nursing students to intervene in elder abuse (Table 4), the higher the awareness of elder abuse, the higher the knowledge of legal and institutional knowledge, and the higher the attitude.

## DISCUSSION

This study was conducted to identify the relationship between the perception of elder abuse, legal and institutional knowledge, the attitude toward elder abuse, and the intention of elder abuse intervention, and to identify factors that affect the intention of intervention based on the results of the study. Most of the prior studies on Korean college nursing students focused on knowledge and attitudes toward older people, and there were no

**Table 4  Factors affecting intention of intervening toward elder abuse.** Factors influencing the intention of nursing college students to intervene in elder abuse.

| Predictors | B | SE | β | t | p |
|---|---|---|---|---|---|
| Elder abuse awareness | 2.479 | 1.122 | .196 | 2.575 | .011 |
| Legal and institutional knowledge of elder abuse | .054 | .027 | .156 | 2.029 | .044 |
| Elder abuse attitude | .068 | .026 | .166 | 2.210 | .029 |
| Sex[+] | .684 | .580 | .102 | 1.178 | .241 |
| Education course about elder abuse[a] | 2.490 | 1.173 | .166 | 2.122 | .035 |
| $R^2 = .129$, $Adj\ R^2 = 1.2$, $F = 4.349$, $p = .001$. | | | | | |

**Notes.**
[a] Dummy variable: Sex (men=0), Education (none=0).

studies on the perception and attitude toward elder abuse and intention of intervention. It was difficult to compare the results of this study with those of the studies on nurses because of the difference in age and knowledge of nursing students, but the purpose of this study was to increase awareness of elder abuse, legal and institutional knowledge, and contribute to elder abuse intervention. Prior research has found that the perceptions of elder abuse and elder abuse were significantly related (*Ko, 2010*; *Park & Youn, 2001*; *Lee & Lee, 2007*; *Malley-Morrison, You & Mills, 2000*). In this study, the average recognition of elder abuse was 3.76 points. This is higher than the 3.07 score of Ko's (2010) study, which was conducted using a five-point scale targeting five general hospital nurses in 2010, and higher than the 3.56 points in *Cho*'s (*2014*) (*Seong et al., 2016*), where a 4-point scale was used for nurses in a senior university hospital (*Seong et al., 2016*; *Ko, 2010*). In this study, legal and institutional knowledge was 5.92 points, and in related studies, emergency room medical personnel often did not know the actual method of reporting, even if they recognized that they had a reporting obligation (*Seong et al., 2016*). Teaching and acquiring accurate definitions of elder abuse, screening of elder abuse, as well as relevant legal procedures and reporting methods, should be included in a future-oriented strategy to improve nurses' sensitivity and knowledge of elder abuse, and ultimately improve reporting rates.

In this study, the attitude toward elder abuse was 2.49 points, lower than the 2.82 points of *Cho*'s (*2014*) study (*Seong et al., 2016*). The lower score could be explained by the study subjects being nursing students, who do not have experience in caring for older people.

Furthermore, the intention to intervene in elder abuse was 3.31 out of 4, indicating a middle or higher level. The results of the degree of abuse perception indicated that physical abuse was recognized as most severe among the various types of abuse.

Regarding emotional abuse, the severity of abuse was relatively low. It was confirmed that nursing students only recognized cases that could be clearly judged externally as elder abuse, such as physical abuse. This implies a need for education in recognizing various types of elder abuse, such as emotional, economic, neglect, and abandonment.

In this study, the perception of elder abuse, legal and institutional knowledge, elder abuse attitude, and intervention intention all revealed significant positive correlations, being consistent with the results of previous studies targeting nurses (*Ko, 2010*; *Park, Choi & Lee, 2013*; *Kim, 2010*; *Song, 2007*). According to the regression analysis, the factors affecting the intention of intervention in elder abuse were identified as variables related
to education, awareness of elder abuse, legal and institutional knowledge, and elder abuse attitudes, with an explanation of 12.9%. This was similar to the results of *Cho (2014)* and *Park & Youn (2001)*, who studied nurses' intention to, and behavior of, reporting child abuse (*Seong et al., 2016*; *Park, Choi & Lee, 2013*). When attitudes are formed, reactions may exist before meeting the object and affect behavior (*Erwin, 2001*). Even in previous studies on knowledge, attitudes, and behaviors regarding older people, knowledge and attitudes influenced behavior (*Kim, Oh & Wang, 2016*).

In Korea, nurses are already obligated to report elder abuse, and if they do not report, they are subject to legal intervention in elder abuse in terms of a fine. If this will be transmitted to the nurses, the community, and the family, the professional responsibility of the nurse to report abuse will increase along with the intervention behavior. To do this, it is necessary to advise the role of nurses as advocates for the underprivileged through education and to share awareness.

This study found that to ensure active intervention when encountering elder abuse, it is necessary to raise awareness, improve legal and institutional knowledge, and investigate attitudes toward elder abuse that affect the intention to intervene; training in geriatric nursing should hence be given. This study proposes further research on the development of educational programs for nursing students to increase their intention to intervene. It is necessary to receive a thorough education on elder abuse in order to recognize abuse cases; the development of a curriculum for this is urgently required.

As with child abuse, it is necessary to educate people on effective recognition methods elder abuse, such as education through role play, and a case-based training approach (*Kim & Lee, 2013*). Furthermore, a specific and systematic procedure and plan should be developed to enable nursing students to easily intervene in elder abuse (*Rosen et al., 2018*).

## CONCLUSIONS

This study was conducted in the absence of research on the perceptions and attitudes of nursing college students regarding elder abuse. This study tried to identify the impact of perceptions of elder abuse, legal and institutional knowledge, and the attitude of elder abuse on the intention of intervention in elder abuse. According to this study, the variables that affect the intention of nursing college students to intervene in elder abuse were education, awareness of elder abuse, legal and institutional knowledge, and attitude. Therefore, to increase the intention of elder abuse interventions in prospective nurses, it is necessary to improve the perception of elder abuse and attitudes toward elder abuse to enhance the discovery and judgment of elder abuse cases and to improve knowledge.

The limitations of this research are as follows:

First, this study was conducted only for nursing college students attending three universities in Chungbuk and Gangwon. Therefore, it is difficult to generalize the results. Repeated studies are required for nursing students in various regions. Second, among the measurement tools used in this study, the legal and institutional knowledge and attitude tools for elder abuse are modified and need to be developed and studied in the future.

Based on the results of this study, the following suggestions are proposed. First, we propose the development and application of education programs for nursing college

students to improve the intention of intervention in elder abuse. Second, we propose a tool development study that is necessary to study the intentions of elder abuse.

### Funding

This work was supported by the Ministry of Education of the Republic of Korea and the National Research Foundation of Korea (NRF-2020S1A5A8046754). The funders had no role in study design, data collection and analysis, decision to publish, or preparation of the manuscript.

### Grant Disclosures

The following grant information was disclosed by the authors:
Ministry of Education of the Republic of Korea.
National Research Foundation of Korea: NRF-2020S1A5A8046754.

### Competing Interests

The authors declare there are no competing interests.

### Author Contributions

- Hee-Jeong Kim and Min-Sook Seo conceived and designed the experiments, performed the experiments, prepared figures and/or tables, authored or reviewed drafts of the paper, and approved the final draft.
- Dahye Park conceived and designed the experiments, performed the experiments, analyzed the data, prepared figures and/or tables, authored or reviewed drafts of the paper, and approved the final draft.

### Human Ethics

The following information was supplied relating to ethical approvals (i.e., approving body and any reference numbers):

Ethical approval (IRB No: 1041479-HR-202005-004) was obtained from the Research Ethics Committee of Namseoul University.

### Data Availability

Raw measurements are available in the Supplemental Files.

### Supplemental Information

Supplemental information for this article can be found online at http://dx.doi.org/10.7717/peerj.12079#supplemental-information.

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
