# Peer review of "Factors influencing intention to intervene in elder abuse among nursing students"

_PeerJ, doi:10.7717/peerj.12079_

## Round 0.1 · original submission · Major Revisions

· Academic Editor

Major Revisions

The authors make some useful suggestions for revising the paper which will improve its quality.

Reviewer 1 ·

Basic reporting

This manuscript reports on a study attempting a Factors Influencing Intention to Intervene in Elder 2 Abuse among Nursing Students. In this respect I applaud the authors for their worthwhile undertaking which reflects a recent spate of research in Korea.
The article is written in a clear and unambiguous way and literature references is enough provided. The article keeps a suitable structure, figures and tables. The literature is connected with the objectives of the research.
However, I would like to make some suggestions to the authors. For example, the authors include the term elderly along the article, but this term is generally considered ageist and stereotyped because it does not take into consideration the heterogeneity of older people which increases with age, for this reason I highly recommend use “older people” or “older adults”. Moreover, it is recommended not use like keywords terms included in the title. In addition, they use the term gender, but I am not sure whether they talk about a social variable (gender) or a biological variable (sex). Maybe they may think about it.

Experimental design

The article define the research questions and they are relevant and meaningful because of it is a knowledge gap about this topic in their geographical context. It seems that the investigation has been conducted rigorously. Although I would like to know more about the sample recruitment method.
As regards the measures, the usual procedure to describe them is to begin with sociodemographic data and after that the other variables.
I would like to know why they adapt two tools developed previously for child abuse to suit with older people. There are some tools that assess elder abuse attitudes for example but they do not used them and maybe for these reason the Cronbach´s alfa is .68.

Validity of the findings

It seems that the results are important because their novelty in their geographical context and they have benefit to literature in this field.
Although I am curious about why they collect some variables, included in table 1, but after that, they do not used them in order to analyze their relationship with the other variables. I think these results could be pretty interesting to future intervention programs.
Furthermore, they include the exact p value when generally the right way is p> o < .01 or .05.
Finally, although the discussion and conclusion section are rather well stated, on the one hand, it would be desirable to make deeply inferences about the controversy between their results and the literature. On the other hand the authors points out general alternatives but it seems like other articles that tackle with this topic. I highly recommended offer more specific practical purposes. I am very interested in the practical application of these results and the contributions to the field. I encourage to authors to include more information about it.
Overall, although I find the study´s findings very valuable, the paper needs more work in order to be of a suitable standard for publication to improve its significance and to increase its appeal or impact on the broader scientific readership in the field. The aim of these comments is only to improve the quality of the article.

Reviewer 2 ·

Basic reporting

No comment

Experimental design

No comment

Validity of the findings

No comment

Additional comments

This research aims to describe the factors influencing the intention to intervene in elder abuse in nursing students. Elder abuse is a very relevant topic. Unfortunately, there is a lack of training in sociosanitary professionals. Assessing nursing students is of special interest to develop specific education programs and guidelines focused on elder abuse prevention and promotion of good practices/care towards older people in long-term care.

The literature review is well-developed and referenced. The methodology is conducted correctly and in enough detail to replicate. Results are well described, statistically sound, including tables that help to follow the main results. Finally, both the discussion and the conclusions are well-stated from the results.

I suggest addressing some minor issues to improve the manuscript.

1. In the introduction, it is clear the importance of evaluating elder abuse in nursing students. However, previous literature on nursing students´ attitudes, perceptions and elder abuse associated factors are not described. The authors state that few studies address this issue, and that's the relevance of developing this study. However, it would be convenient to include previous articles developed although they are only a few.

2. The results show that nursing students are more able to detect physical abuse. Considering this, you suggest developing training in also other types of abuse, which is correct. I would also discuss further in the discussion section the importance of training nursing students on implicit types of abuse since it seems they detect better explicit types. For example, infantilization (infantilized communication, not empowering older people, etc) is a type of violence towards older adults very related to negative stereotypes and most professionals are not conscious they can be disrespecting and emotionally hurting the older adult. Universities and educational courses could also work on negative stereotypes towards older people, empowerment, recognition of their dignity, etc, in order to prevent elder abuse.

3. The English language can be understood clearly but some sentences could be improved. I include two examples below. However, I would review the complete text.
- “The intention to intervene in elder abuse was the perception of elder abuse (r=.26, p=.005), legal 215 and institutional knowledge (r=.54, p=.046), attitude (r=.121, p=.004), and there was a 216 significant positive correlation”. (lines 214-216). I would replace it with “The intention to intervene showed a significant positive correlation with the perception of elder abuse (r=.26, p=.005), legal 215 and institutional knowledge (r=.54, p=.046) and attitude (r=.121, p=.004).”
- "The descriptive research design was conducted by completed the questionnaires to 182 nursing students." Reformulate this sentence.

Reviewer 3 ·

Basic reporting

The English language needs improvement to ensure readability by an international audience. The following lines include examples of where language needs revision: 47-52, 60, 65, 67, 72, 79, 93-94, 155-156, 214, 269-270, 272-273, 274, 287-288, 295-296. I recommend the authors use an editor or colleague who is proficient in English to review the wording.

The introduction and background need to more clearly indicate the gap in evidence that this study addresses. The authors do a good job addressing an increase in elder abuse, nurses' need to report abuse and factors associated with medical providers intention to intervene. However, the authors are missing evidence that explains and connects this healthcare issue with nursing education. For example, the statement on line 74 regarding research with nursing students is not supported by evidence. I am unclear why the authors chose to address nursing education instead of nurses based on this review of literature. Lines 250-252 with supportive literature would be useful to include in the background section to substantiate the need for this study.

Examination or description of the coursework that students take in nursing programs would be useful to explain what content students are regulated to receive by agencies that have oversight for nursing education.

Thank you for including Tables 1 and 2. Please include more content to clarify these tables. For example, in Table 1, explain to what "differences in intention of intervening" refers. The use of the word 'difference' appears to suggest differences within the sample and the statistics reported do not support an evaluation of differences between participants. The authors could simply name this table as 'sample characteristics'. Further, clarify variable names in Table 1. For example, explain to what 'need for education' refers. Data analysis could be improved by including confidence intervals for results in Table 2.

Tables 3 and 4 are thoroughly done.

Thank you for providing raw data. However, supplemental files need more descriptive metadata identifiers to be inclusive of international audiences.

Experimental design

This paper reports an original research study within the aims and scope of the journal. Elder abuse is an important public health issue.

The authors indicate an increase in elder abuse and the need to address this issue among healthcare providers. The author could provide more evidence to justify the need to address education of nurses and nursing students regarding elder abuse. For example, the connection made by the authors between fewer reported cases of abuse and lower perceptions, intention or knowledge regarding elder abuse among nurses needs to be substantiated more clearly with evidence.

Authors did a thorough job explaining the use of factors on lines 67-70 to substantiate why these were included as predictors and an outcome variable.

Authors need to provide more detail about the methods to allow replication of this study. Specifically, the authors need to explain how students were recruited and data was collected. Authors needs to explain how coercion of students was minimized. For example, explaining the relationship of the authors to the students would be useful to explain how students voluntarily agreed to participate and if faculty were connected to the research study.

Explanation if the Crohnbach alpha values for each instrument were acceptable would increase validity of the use of these instruments in this study.

The sample needs to be described to be inclusive of international audiences. For example, on line 189, 'the fourth grade' refers to individuals 9 to 10 years of age in the United States. Further, levels of degree preparation need to be described in a way that is inclusive of international audiences.

Validity of the findings

Thank you for providing raw data. However, supplemental files need more descriptive metadata identifiers to be inclusive of international audiences. I have no comment about this data for this reason.

Authors do a thorough job of comparing the study findings to other evidence on lines 255-264.

Wording on lines 269-277 could more clearly supported by the data from the study. For example, authors could explain how nursing students do not take care of elderly (line 270). Perhaps the authors mean that students do not have as much experience taking care of elderly. Additionally on line 272, authors could state that participants indicated that of all types of abuse, physical abuse was rated as the most severe and explain what 'most severe' means.

The conclusions on lines 292-305 appear to overreach the results. The study results suggest that certain factors are associated with intention to address situations of elder abuse. The link between intention to intervene and actual intervention in situations is not clearly substantiated by evidence. Authors could indicate that these factors were associated with intention in this study and further exploration between intention and behavior needs to be established.

Additional comments

Thank you for addressing an important public health issue. The authors' most important issue is to improve the English language in the paper and in the tables. The next important issue is adding evidence to support the rationale for this study. The next most important issue is to add detail to the methods section. A final point would be to provide resources for the reader would add to this paper. For example, the authors provide examples of resources for more information about abuse, reporting standards, and case studies or simulation exercises that could increase reader proficiency in addressing this issue.

I hope this review is helpful. This is an important issue and I hope you will continue this work. Thank you for inviting me to review this paper.

---

## Round 0.2 · Minor Revisions

· Academic Editor

Minor Revisions

I have sent your revised paper back to the original reviewers. One recommends acceptance, while the other suggests a minor change in your terminology, from 'older people abuse' to 'elder abuse'.

I will leave it to you to decide which term you prefer, but I agree with the reviewer that 'elder abuse' sounds better in English.

Reviewer 1 ·

Basic reporting

I would like to thank to the authors for the revision of the article.
The term elder abuse is correct, only in this circusntance elder could be used because it is the consensuous term to refer abuse towards older people.

Experimental design

The authors make the changes requested.

Validity of the findings

The authors make the changes requested.

Additional comments

The authors make the changes requested.

Reviewer 2 ·

Basic reporting

no comment

Experimental design

no comment

Validity of the findings

no comment

Additional comments

I do think that my concerns have been addressed in this version. The methodology is sound and offers interesting findings. Having done the changes suggested, I believe that this important piece of research is ready for the next stage of the publishing process.
Thank you for your work!

---

## Round 0.3 · accepted · Accept

· Academic Editor

Accept

Thank for the additional edit.